# Cartilaginous Intrusion of the Atrioventricular Node in a Quarter Horse with a High Burden of Second-Degree AV Block and Collapse: A Case Report

**DOI:** 10.3390/ani12212915

**Published:** 2022-10-24

**Authors:** Sarah Dalgas Nissen, Arnela Saljic, Sofie Troest Kjeldsen, Thomas Jespersen, Charlotte Hopster-Iversen, Rikke Buhl

**Affiliations:** 1Department of Veterinary Clinical Sciences, Faculty of Health and Medical Sciences, University of Copenhagen, Højbakkegaard Allé 5, 2630 Taastrup, Denmark; 2Institute of Pharmacology, West German Heart and Vascular Centre, University of Duisburg-Essen, 45117 Essen, Germany; 3Department of Biomedical Sciences, Faculty of Health and Medical Sciences, University of Copenhagen, 2200 Copenhagen, Denmark

**Keywords:** equine, syncope, cardiac arrhythmias, implantable loop recorder, AV nodal histology, cardiac cartilage, His bundle

## Abstract

**Simple Summary:**

The atrioventricular node is fundamental in securing electrical conduction in the heart. In horses, blocked atrioventricular conduction, known as second-degree atrioventricular block, is a common finding and considered harmless if only occurring during periods of rest. However, some horses may experience low blood pressure if conduction is very slow, potentially resulting in syncope and collapse. In this presented case, a Quarter horse was referred to the hospital at the University of Copenhagen as it suffered from multiple collapses at rest. The clinical examination, including surface electrocardiography and long-term cardiac monitoring with a subcutaneously implanted monitoring device, revealed a pronounced number of second-degree atrioventricular blocks and low ventricular rate. Following euthanasia, investigation of the atrioventricular node revealed severe cartilaginous changes of the area around the aortic valve, which was intruding into the atrioventricular node in the His bundle region. Atrioventricular nodal abnormalities, such as fibrosis formation, have been suggested to interfere with atrioventricular conduction in other horses; however, a specific diagnosis could not be established in this specific case and more knowledge on atrioventricular nodal disease in horses is required.

**Abstract:**

Second-degree atrioventricular (AV) block is the most common cardiac arrhythmia in horses, affecting 40–90% depending on breed. Usually, the AV blocks occur while the horses are resting and disappear upon exercise and are, therefore, considered to be uneventful for horses. However, if the AV blocks occur frequently, this may result in syncope and collapse. Identifying the cause of second-degree AV block is difficult and often subscribed to high vagal tone. In this report, we present an eight-year-old Quarter horse with a high burden of second-degree AV blocks and multiple collapses. The clinical examination, including neurological examination, blood analysis, 24-h ECG recording and cardiac echocardiography, did not reveal any signs of general or cardiovascular disease besides a high burden of second-degree AV blocks (~300 blocks per hour) and a hyperechoic area in the AV nodal region. An implantable loop recorder (ILR) was inserted to monitor the cardiac rhythm. The ILR detected several consecutive second-degree AV blocks and pauses above 5 s. However, unfortunately, no recordings were available during the collapses. Eventually, the horse was euthanized and the heart inspected. The aortic root was severely cartilaginous and appeared to penetrate the AV node, especially in the His bundle region, possibly explaining the hampered AV conduction. Nevertheless, it is still uncertain if the AV nodal disruption caused the collapses and more knowledge on AV nodal diseases in horses is warranted.

## 1. Introduction

Second-degree atrioventricular (AV) block is the most common cardiac arrhythmia among horses, with a reported prevalence of up to 40–90% depending on breed [1,2]. Most often, the arrhythmia does not inflict any clinical impact on the horses and is deemed harmless if the second-degree AV blocks disappear when vagal activity is withdrawn, such as under light exercise [3]. Most clinical cases covering the AV nodal function report third-degree AV block as a result of severe systemic disease or local damage of the AV node [4,5,6,7], but little is known about the etiology of second-degree AV blocks. Only a few studies of pronounced second-degree AV blocks have reported associations to inflammation or degenerative changes of the AV node [8,9,10]. The potential pathological impact of second-degree AV block is most likely unnoticed, as only a few studies have investigated this matter. Furthermore, as we currently lack diagnostic modalities, the clinical examination may, therefore, not include a thorough inspection of the AV nodal function in the horse. Here, we present a case of a Quarter horse presenting with multiple episodes of collapse and a subsequent finding of abnormal cartilaginous intrusion within the AV node and His bundle, diagnosed postmortem.

## 2. Case Presentation

An eight-year-old Quarter horse mare (474 kg) was referred to the Large Animal Hospital at the University of Copenhagen in November 2018, for evaluation of multiple collapse episodes during rest. The first collapse had been observed at the farrier, but as the horse, over a one-year period, had several unexplained injuries on the extremities, several unobserved collapses during nighttime were suspected by the owner.

### 2.1. Clinical Investigation at Referral Center

On physical examination, the horse was alert and responsive. Rectal temperature, respiratory rate and heart rate were within normal references. Four clear heart sounds were identified upon cardiac auscultation with no murmurs, but with numerous pauses occurring with regular intervals. Diffuse edema was observed in both hind legs. The edema was not present at subsequent examinations. The clinical neurology examination did not reveal any noticeable abnormalities and no additional neurological examinations were performed. Hematology, biochemistry and electrolytes were all within normal references.

A 24-h Holter ECG (Televet100, KRUTECH Televet, Kruuse A/S, Maarslev, Denmark) confirmed the presence of multiple second-degree AV blocks and a heart rate ~40 beats/min when second-degree AV blocks were absent and 25–30 bpm when second-degree AV blocks were present. The ECG revealed episodes of ~300–400 s-degree AV blocks per hour, PR interval ~450 ms (range 380–600 ms), PP interval ~1400 ms and normal QRS complexes (Figure 1). The PR interval varied greatly towards a block showing incremental properties most often, with the second last or last preblock PR interval being the longest. A second-degree AV block with nonconsistent PR intervals is comparable to the Mobitz type I AV block in humans, also known as Wenckebach AV block [11], which is characterized by a progressive prolongation of PR intervals towards the blocked beat. The second-degree AV blocks vanished in the horse when the heart rate increased above approximately 50 beats/min. An ECG during a 20-min exercise test in trot and canter revealed no abnormalities. Postexercise, the second-degree AV block reoccurred shortly after recovery, when the heart rate decreased to 60 beats/min. Heart rate variability (HRV) analysis was performed on a 10-h resting ECG recording using software for HRV analysis (Kubios HRV version 3.5 for Windows, MATLAB, The MathWorks Inc., Kuopio, Finland) with and without applying a filter enabling the exclusion of second-degree AV blocks, according to Eggensperger and Schwarzwald (2017) [12]. Without a filter, the mean normal-to-normal intervals (MeanNN) corresponding to the corrected RR intervals was 1574 ms, the Standard deviation of NN intervals (SDNN) was 839.3 ms, the standard deviation quantifying the dispersion of data points in a Poincaré plot perpendicular to the line of identity (SD1), a measure of short-term variability, was 593.5 ms and the standard deviation quantifying the dispersion of data points in a Poincaré plot along the line of identity (SD2), a measure of long-term variability, was 552.3 ms; all being high values most likely due to the many second-degree AV blocks. After applying automatic filtering excluding all second-degree AV blocks, the MeanNN was 1363 ms, the SDNN was 112.4 ms, the SD1 was 57.9 ms and the SD2 was 148.1 ms; all within or below previously reported ranges for normal horses, indicating that this horse had normal-to-low sinus beat-to-beat variability.

Echocardiography was performed with a portable Vivid IQ ultrasound system (GE Healthcare, Chicago, IL, USA) with a 1.3–4.0 MHz phased array transducer and showed a normal heart size and function. An intermittent trivial (nonaudible) mitral valve regurgitation was visible with color Doppler echocardiography. In the right parasternal, long-axis, four-chamber view of the heart, a slightly hyperechoic area (~1.5 × 0.8 cm) in the AV nodal region was observed (Figure 2). An implantable loop recorder (ILR, Reveal LINQ, Medtronic, Minneapolis, MI, USA) was inserted subcutaneously in the 5th intercostal space, as previously described [13], in order to monitor the cardiac rhythm further upon discharge. The ILR was programmed to detect pauses above 4.5 sec in duration.

### 2.2. Monitoring Period after Discharge

The ILR was investigated upon discharge from the hospital in November 2018 and again, one month later, in December 2018. During the winter, no collapses were observed, although several unexplainable injuries on the horse were observed in the mornings, indicating collapses while the horse was left on its own. In the following summer (June and August 2019), two more episodes of collapse were observed by the owner. One episode occurred while the owner was sitting on the horse after a ride and the second episode occurred while the horse was standing in a trailer. Common for all three collapses observed by the owner was that the horse dropped down on its side without any warning. After less than one minute, the horse got up again and appeared bright, with no sign of blindness and was responsive. Upon investigation after the last collapse in August 2019, the ILR, unfortunately, had ran out of battery and no episodes were, therefore, recorded. A new ILR was inserted in October 2019 and was investigated one and two months later. The ILR revealed the constant presence of second-degree AV blocks and, at each investigation, numerous episodes where two consecutive second-degree AV blocks appeared in a row, generating pauses of more than >5 s of duration (Figure 3). Often, the pauses were only interrupted by two conducted beats followed by two consecutive second-degree AV blocks again. The heart rate varied between 20–30 bpm in the ILR recordings. As treatment options were limited, and the owner felt unsafe to be near or ride the horse, the owner decided to have the horse euthanized in November 2019.

### 2.3. Autonomic Nervous System

Before euthanasia, the owner gave consent to conduct a pharmacologic blockade of the autonomic nervous system, in order to explore the pathogenesis of the second-degree AV blocks further. The horse was equipped with a modified base apex surface ECG. Blockage of the parasympathetic nervous system was conducted by the infusion of atropine (Atropine sulphate, 10 mg/mL; 0.04 mg/kg) and the sympathetic nervous system was blocked by the infusion of propranolol (Propranolol hydrochloride, 10 mg/mL; 0.2 mg/kg) through an intravenous catheter, as previously described [14,15]. Approximately two minutes after the infusion, the heart rate stabilized at 65–70 beats/min and the second-degree AV blocks disappeared in the 30-min monitoring period.

### 2.4. Macroscopic and Microscopic Tissue Investigations

Following euthanasia with a lethal intravenous injection of pentobarbital (Euthasol^®^Vet, Produlab Parma B.V., Raamsdonksveer, The Netherlands 400 mg/mL; 140 mg/kg), the heart was taken out and flushed with cardioplegi. Upon macroscopic inspection of the heart, no abnormalities were discovered, besides an abnormal area of 1 × 0.5 cm in the AV nodal region at the aortic root, where the base of the noncoronary aortic leaflet was severely cartilaginous. Cardiac cartilage is common in horses [16]; however, in this case, the structure was covering the AV node in the area of the penetrating bundle (Figure 4). For the collection of the AV node, the Triangle of Koch was identified in the right atrium, and a sample from the tip of the triangle of approximately 3 × 3 cm was collected and immediately frozen in a mixture of isopentane and dry ice, as previously described [14]. The sample was stored at −80 °C until further examination. Next, the AV nodal sample was cryosectioned perpendicular to the AV node in sections of 30 µm and collected onto glass slides (Superfrost plus, VWR International ltd, Radnor, PE, USA). For every seven collected samples, 450 µm of tissue was discarded in order to collect samples throughout the entire AV node. Histologic staining with Picro Sirius Red staining was performed to visualize the component structures of the AV node and level of fibrosis. The slides were imaged using the Zeiss Axio Scan.Z1 Slide scanner at 10× magnification. For comparison, the AV nodes from 17 healthy retired Standardbred racehorses (mean age: 6.8 ± 2 years), previously harvested for another cardiac study, were sectioned and stained in the same manner (Figure 4). The control horses were research animals (approved by the Danish Animal Experiments Inspectorate, license number 2016-15-0201-01128 and by the local ethical committee at the Department of Veterinary Clinical Sciences, University of Copenhagen). Results from 24-h Holter ECG, echocardiography, blood hematology and biochemistry from the control horses were all within normal ranges and with median second-degree AV blocks of 25 (range 0–116) per hour from the 24-h Holter ECG.

### 2.5. Histological Results

The histology of the consecutive sections of the AV node confirmed the presence of a cartilaginous structure at the base of the aortic leaflet in conjunction with the AV node and the penetrating bundle, also known as the His bundle (Figure 4). An automatic quantification method of the amount of collagen [17] revealed no obvious differences in the amount of collagen, with 83% collagen in the case horse and a mean of 82 ± 17% in the control horses. The major difference was the location of the AV node in conjunction with the cartilage. For all the control horses, the AV node and the cartilage were separated throughout the entire AV node, whereas for the case horse, the cartilage covered the AV node in the transition into the His bundle (Figure 4 and Appendix A).

Furthermore, the maximal diameter and length (0.5 × 1.3 cm) of the cartilage structure was larger in the case horse compared to the control horses (0.5 × 0.5 cm).

## 3. Discussion

Here, we report a clinical case presenting with multiple collapses without any apparent signs of disease. However, the horse experienced a high burden of second-degree AV blocks, accompanied by severe cartilaginous changes of the basal aortic leaflet, intruding the AV node and His bundle. Calcification of the mitral and/or the aortic valve, and fibrotic formation of the AV node and His bundle have also been described to induce second-degree AV blocks in humans and dogs; however, to the authors’ knowledge, no reports on cartilaginous changes in other species have been described [18,19,20,21]. The horse in the current case did not suffer from complete AV block and conduction through the AV node could be restored by blocking the muscarinic acetylcholine receptor with atropine. In human patients with second-degree AV block, even in cases with idiopathic fibrosis, myocardial ischemia or inflammatory conditions, atropine is often used to restore AV node/proximal His conduction and is recommended as the first drug of choice in symptomatic bradycardic patients [22]. It is unclear why blocking of the parasympathetic nervous system can rescue AV conduction disturbances caused by ischemic conditions or fibrosis, but perhaps changing the threshold level for cellular depolarization, as atropine does [23], is adequate to overcome the conduction slowing induced by calcification or fibrotic infiltration. The horse was more sensitive to parasympathetic stimulation indicated by the high burden of second-degree AV blocks compared to other horses during resting periods [14], especially when considering the ILR recordings that revealed multiple consecutive second-degree AV blocks. Despite not having complete heart block, also known as third-degree AV block where the atrial rhythm is dissociated from the ventricular rhythm [11], the very low heart rate as a result of multiple second-degree AV blocks resulting in longer ventricular pauses, may explain why the horse experienced multiple collapses. For some horses, pronounced second-degree AV block is appropriate, but for this particular case it might be that the very low heart rate during the AV blocks affected the horse clinically [9]. The blocks occurred during rest or in the transition from activity to resting periods, where a surge of parasympathetic activity is present. A single pause exceeding three–four seconds is not unusual for a horse [4] and it is uncertain whether multiple pauses may have symptomatic effects. Nonetheless, none of the above mentioned diagnostic modalities nor the histologic findings in the AV node confidently identify the reason for the collapses observed in the presented case. Several diseases have been reported to result in collapses, but often it can be difficult to identify the cause of collapse. In a retrospective study by Lyle et al., a final diagnosis was obtained in only 11/25 horses and, of these, 4/11 were of cardiovascular origin [24]; however, second-degree AV block was not the only finding in these cases. Additionally in this study, ILRs were used in order to attain long-term monitoring which has been reported to increase the likelihood of identifying the origin of collapses of unknown reason in humans [25]. Lyle et al. suspected syncope of neurocardiogenic origin in one horse with an ILR which had stored an ECG during two collapses. The collapses were not associated with any cardiac arrhythmias [24]. In the current case, the ILR added information about the severity of the second-degree AV blocks in the Quarter horse that was not identified during the 24-h Holter ECG recording. However, no recordings during collapses were obtained due to the lack of battery in the ILR. In future cases, it is advocated to investigate and extract data from the ILR more often, especially right after a collapse. Additionally, video monitoring of the horse at night might elucidate how often collapses actually occurred and, combined with Holter monitoring, this could have provided insight into the causes and frequency of collapses in the current case. Currently, we do not know if second-degree AV block can potentially result in collapses in horses. For this specific case, other explanations are as reasonable, such as sleep deprivation, narcolepsy, hypoglycemia or neural syncope [24]. A mitral valve regurgitation was also identified in the present case; however, as the chamber dimensions were within normal ranges, and no valvular or cordal thickening was present, the regurgitation was considered mild and unlikely to result in collapses [26].

Postmortem evaluation and histology of the AV node in horses with second- and third-degree AV block are rare but have previously been reported [9,10,27]. In these cases, one horse with third-degree AV block was diagnosed with fibrous calcification of the AV node; however, this horse did not respond to atropine and the calcification was related to the AV node itself and not the surrounding tissue [27]. Another horse with advanced second-degree AV block was diagnosed with inflammation of the AV node and the bundle of His [10]. However, all of these cases presented without detailed information of the histology. Other studies reporting on the histologic characteristics of the AV node in healthy horses exist, in which some mention the presence of cardiac cartilage or even a cardiac bone [28,29,30,31]. The aortic root lies in the continuation of the central fibrous body, which in many mammals is transformed into cartilage and referred to as the cardiac cartilage [31]. Cardiac cartilage is common in horses [16]; nonetheless, a precise description of the cardiac cartilage is sparse and it is, therefore, uncertain how much it varies in size and how much cartilage is normal in horses. Likewise, it is uncertain whether the cartilage is able to disrupt the electrical conduction through the AV node. The authors of the present study have carefully dissected and examined the AV nodal area of 30 horses, both macroscopically and histologically. In total, 15 of these horses presented with cartilage in the AV nodal area; however, to a markedly lesser degree than the case reported here and often also less than the control horses in this case (personal observation).

In humans, second-degree AV block (Mobitz type I) is thought to occur at the level of the AV node, where conduction progressively prolongs and may be associated with ischemia, myocarditis, postcardiac surgery, increased vagal tone or medications that slow AV nodal conduction [11]. If symptomatic, these patients most often respond well to treatment with atropine and rarely need a permanent pacemaker. This case also responded to atropine; however, long-term treatment with atropine is unrealistic in equine medicine, due to the side-effects. We did not include additional electrophysiological testing of the AV node function, although it would have been interesting to explore at which level the AV nodal blocking occurred. It is unknown if second-degree AV block with progressive prolongation of the PR interval also occurs in the AV node as in humans. We recently reported His signals from a case with second-degree AV block where the PR interval varied towards blocking but the His signals were present in the blocked beats, suggestive of infra-Hisian blocks [2]. This is an interesting observation, but more studies are needed to clarify this.

In people where the AV nodal conduction tract is damaged or mechanically affected by aortic valve surgery, pacemaker implantation may be needed [32,33,34]. The implantation of a permanent pacemaker for therapeutic purposes in various brady-arrhythmias is now also an option in horses and could potentially have been applied in this case as well [35,36]. In humans, the close proximity of the subaortic outflow tract to the AV node makes aortic damaging a possible contributor to both intermittent and complete heart block [32,37], where also a difference in AV nodal anatomy may increase the susceptibility to AV nodal injury [33]. This may also be the case for horses, although to the authors’ knowledge no previous reports exist of cases in horses where cartilage in the subaortic region is so pronounced that it may interfere with cardiac conduction. Even though the echocardiographic examination revealed a hyperechogenic structure in the AV nodal region, this is also observed in horses without a history of collapse or high number of second-degree AV blocks and could not alone explain the collapses [38]. However, in horses with multiple AV blocks, echocardiography could add information about anatomical changes in this region. Though still being a rare procedure and not yet clinically available in horses, intra-cardiac 3D mapping and His bundle recordings could provide information on the electrical conduction in the AV nodal region as well [2,39].

Therefore, the diagnosis in horses relies on postmortem examination of the AV nodal region and treatment options remain limited. Finally, this case highlights the fact that, even though the second-degree AV blocks disappear during exercise, they may still affect the horse at rest and AV nodal disease should not be discarded immediately in such cases.

## 4. Conclusions

Cartilaginous presence in the aortic root may lead to the development of second-degree AV blocks in horses and can be suspected on echocardiography as a hyperechogenic structure. The surface ECG revealed long episodes with an extreme presence of second-degree AV block and the ILR detected multiple consecutive AV blocks, accompanied by a very low heart rate. Often, second-degree AV blocks are considered harmless if they disappear upon exercise and we do not know if second-degree AV blocks may have symptomatic consequences for horses at rest. Therefore, more focus on the origin of AV conduction block and the clinical effects should be pursued and supported by more in-depth studies on AV nodal conduction in horses both ante- and postmortem.

## Figures and Tables

**Figure 1 animals-12-02915-f001:**
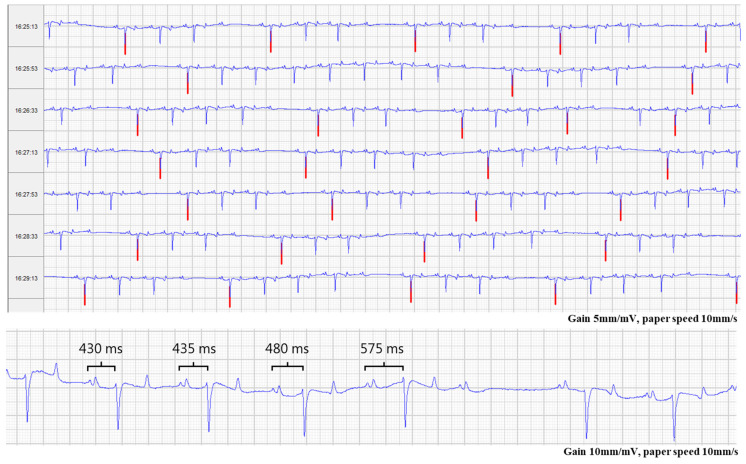
Resting electrocardiogram. Top: Representative electrocardiographic (ECG) strip of four minutes’ continuous recording from the 24-h resting ECG recording. Several second-degree AV blocks were present throughout most of the ECG recording often in patterns of 4:3 in P to QRS ratio. Red lines indicate post AV block beats. Bottom: The PR interval (black clamps) varied in duration prior to AV block most often showing progressive prolongation of the PR interval corresponding to the Mobitz type I block seen in humans. AV: Atrioventricular.

**Figure 2 animals-12-02915-f002:**
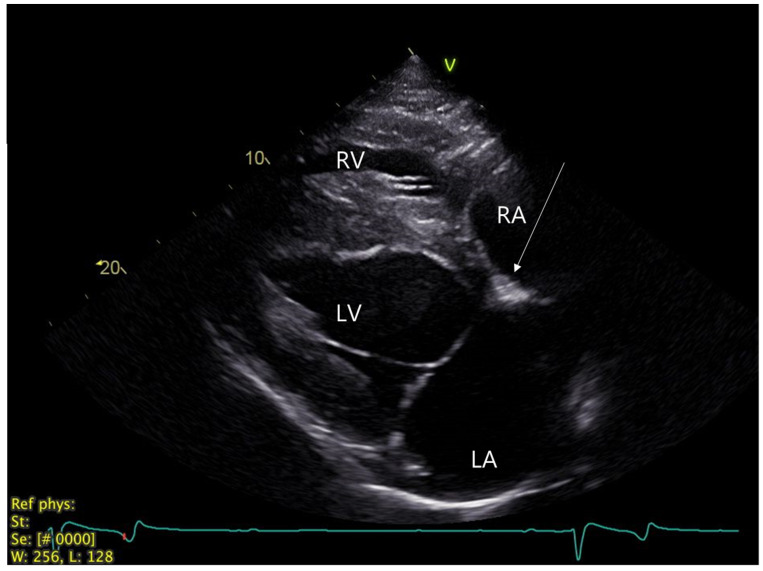
Echocardiogram. Right parasternal, long-axis, four-chamber view of the heart with focus on left atrium. Arrow highlights hyperechogenic zone in the proximal interventricular region (AV nodal area). AV: Atrioventricular, LA: Left atrium, LV: Left ventricle, RA: Right atrium, RV: Right ventricle.

**Figure 3 animals-12-02915-f003:**
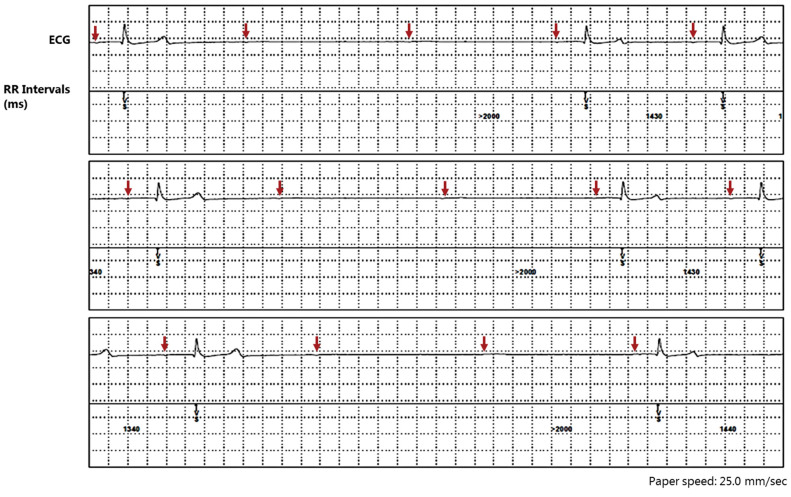
ECG from the implantable loop recorder. Several episodes were detected as pauses by the implantable loop recorder. All stored ECGs from the loop revealed several second-degree AV blocks. Often, two consecutive P waves were blocked. The amplitude of the P waves was very low and is, therefore, pointed out by arrows. The P waves are more readily visible directly in the programmer used to investigate with the loop recorder in order to extract data. Unfortunately, data were only printed on pdf files and no data were stored on the programmer; therefore, quality and amplitude are low. The above representative is a continuous recording sampled by the loop one afternoon in October 2019. VS: ventricular sensing.

**Figure 4 animals-12-02915-f004:**
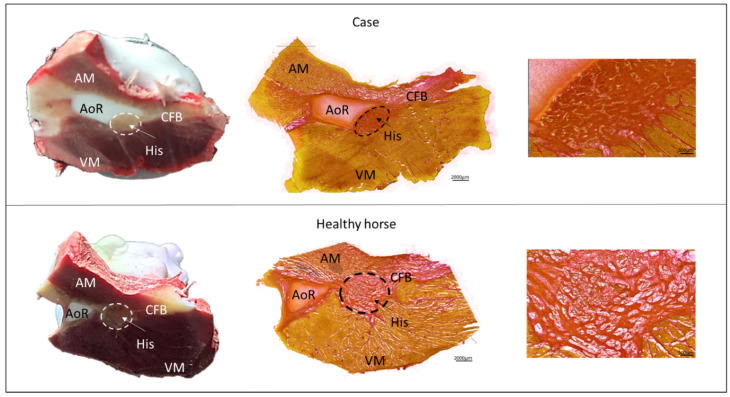
Tissue sample of the atrioventricular node and His bundle. (**Left**): Representative images of transvers sectioning of the AV nodal region from the case and a healthy horse. The very prominent cartilaginous aortic root was already visible macroscopically. (**Middle**): Sirius red histology of the AV node when it transits through the CFB corresponding to the His bundle. Red is collagen whereas yellow is cardiomyocytes. The black encircled area is the His bundle. White areas within the tissue are artifacts induced by freezing of the tissue. (**Right**): Magnified view of the His bundle. Note the very compact bundle and the large cartilaginous part of the AoR in the case horse. AM: Atrial myocardium, AoR: Aortic root, CFB: Central fibrous body, VM: Ventricular myocardium.

## Data Availability

Not applicable.

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
