# Peer review of "Cartilaginous Intrusion of the Atrioventricular Node in a Quarter Horse with a High Burden of Second-Degree AV Block and Collapse: A Case Report"

_animals, 2022, doi:10.3390/ani12212915_

Round 1

Reviewer 1 Report

This manuscript describe a very unusual case of a second-degree AV block in a horse with clinical significance. The literature consider that AV bloscks are benign and without clinical relevance. I recommend to publish to encourage other authors to improve the research in horses with second-degree AV blocks as the authors said in the discussion.

Additional Comments:

What are the main claims of the paper and how significant are they?

This is an unusual case that it have not previously reported in horses. In the literature, second degree AV blocks are benign and not associated with clinical signs.

How does the paper stand out from others in its field?

It is the first time that a horse with an second degree AV block suffer from cardiac insufficiency.

Are the claims novel? If not, which published papers compromise novelty?

Yes, the case is novel.

Are the claims convincing? If not, what further evidence is needed?

The paper is a case report. You can no get strong evidences just with one case.

Are there other experiments or work that would strengthen the paper further?

Authors can include more data about the mitral valve regurgitation (it was audible, the degree of the murmur). Moreover they can include in the discussion why the considerer the cause of the clinical signs the AV block and not the mitral regurgitation.

How much would further work improve it, and how difficult would this be?

The comments are not going to be difficult to be included and can improve the work significantly

Would it take a long time?

Not really. I think it can be solved within a couple of hours.

Are the claims appropriately discussed in the context of previous literature?

Yes, the references are appropriates.

Reviewer 2 Report

This manuscript describes a case of advanced second degree AV block in an 8 year old Quarter Horse. Histopathology of the AV node revealed cartilaginous intrusion of the AV node. Unfortunately the authors could not demonstrate direct association between the clinical signs (collapse) and the bradycardia. This is a major limitation of this case report. Furthermore, additional electrophysiological testing would have been beneficial to evaluate the exact location of the AV block (atrioventricular, infra-Hisian). How do the authors explain the fact that AV node conduction could be restored by atropine? It would be interesting to elaborate more on this fact in the discussion, and to compare this to findings in human patients (e.g. line 264 and further). Finally, the authors should discuss in depth why ILR implantation was preferred instead of longer Holter monitoring with video surveillance, and why pacemaker implantation was not considered.

For histopathology results were compared with a control horse, however, this horse has a different age and breed. In the discussion, the authors describe that they have results for 30 control horses. Why are these results not included in the manuscript and did you choose to include only 1 control horse?

In addition, I have some additional comments and remarks listed below.

Simple summary:

Line 13: ‘concise’ seems like an odd word here

Line 14: ‘slowed’ should be blocked

Abstract: age and breed of the horse should be mentioned

Line 30: ‘some horses may be affected at rest’ – what do you mean here? Clinical signs at rest?

Line 34: you did see a hyperechoic region in the four chamber view

The abstract should also mention whether any events could be recorded with the ILR.

Line 63: ‘in’ November

Line 70: Could the long AV interval be detected by auscultation, since 4 heart sounds were audible?

Line 74: cTnI value?

Line 78: PR interval range 380-500 ms – is this the range for the PR interval measured in this horse? Or the reference range? In Fig. 1 the PR interval is 575 ms?

Figure numbers should be corrected, the numbers in the text do not line up with those in the figure legends.

Line 130: what do you mean by ‘rinsed for blood coagulants’?

Line 159-161: it seems like a contradiction that collagen was very dense but automatic quantification revealed no obvious differences in amount of collagen?

Figure 2 (ILR recording): clarify the duration of the recording shown in the figure. When was this recording made? Are these separate segments or a continuous recording? What do you mean by ‘left’?

The discussion starts with a lot of repetition of the results. I suggest to make this more concise and to include more comparison with the literature (equine, but also human and canine cases).

Line 223: elaborate on other potential causes for collapse in this case. Also elaborate on the previous use of ILRs (line 223-225), what were the diagnoses during collapse?

Line 226-227: The sentence ‘Lyle et al. suspected syncope …’ seems to be grammatically incorrect.

Line 234 and further: It seems appropriate to discuss the case of Luethy et al. here as well?

Line 271: this is shown in the clinical commentary of Decloedt (2021): Cardiac arrhythmias as a potential sign of systemic disease

Reviewer 3 Report

Very interesting data showing a case report of a cartilaginous intrusion of the atrioventricular node in a horse with a high burden of second-degree AV block and correlating with collapse. I believe that major revisions with some edits/suggestions for further consideration could enhance the paper. Some phrases may confuse the reader, leading him to think that the AV Block is the cause of the collapse, which it has not been proven.

Simple Summary

Line 23: In my opinion, it’s not possible to describe: “The cartilage may disrupt the normal electrical activity and may therefore explain the slowed conduction and the periodic collapses” This is an hypothesis that can’t be described in the Simple Summary as it implies that the electrical activity of the tissue was tested and that the differential diagnoses of the causes of collapse in horses were made.  

Abstract

Line 35: “An implantable loop recorder (ILR) was inserted to monitor the cardiac rhythm during events”. The moment of collapse was not correlated with the changes in the ILR. I understood that “events” means “collapse”.

Introduction

Line 57: “Here we present a case of a Quarter horse presenting with multiple episodes of collapse possibly induced by abnormal cartilaginous intrusion within the AV node”. Again, there are other possibilities that were not discussed, for example: hypoglycemia and narcolepsy.  

Case presentation

Line 66: “the horse was 66 observed laying down, so no sleep deprivation disorders were suspected.” This is little information to discard the hypothesis of sleep deprivation. Recumbent horses not necessarily are sleeping. Horses need to achieve REM sleep (about 40 minutes per day) to rest satisfactorily and it’s necessary the muscle atony to reach this stage. Collapses at less busy times of the day and during the night, absence of collapses during the exercise or stressful times and unexplained injuries on the extremities are very common in horses with sleep deprivation. It’s necessary more information to not suspect of it.

Line 72: “Diffuse edema was observed in both hind legs”. Edema is cited only in the first examination. Is it persist during the other periods? It’s common to find this edema in stabled horses that don’t exercise itself. ” The neurological examination did not reveal any noticeable abnormalities”. Have any complementary exam of neurological system been performed? Or only the physical exam of neurological system?

Discussion

Line 218: “if only few conducted beats follow the non-conducted, this may result in hypotension.” This is not measured.

Line 222: “In a retrospective study by Lyle et al. a final diagnosis was obtained in only 222 11/25 horses and of these 4/11 were of cardiovascular origin.” In this study, no case was related only to second degree AV block and it’s not possible to identify if in this specific cases the collapses occurred during the rest.

Conclusion

Line 287: “however the AV block may actually have symptomatic consequences for the horse at rest”. It’s not possible to conclude this, all the symptoms could be other causes.

There is a good description of the histological findings of a large cartilaginous intrusion that may justify a rewrote of the paper with possible relevance to the case report. However, at various times, the relationship between the collapses and the AV block was considered probable, which I would classify as improbable with the data presented.

Another information that may bring relevance to the article is the Heart Rate Variability data, which can be calculated with the ECG performed. I suggest the reference of Eggensperger, 2017 – “Influence of 2nd-degree AV blocks, ECG recording length, and recording time on heart rate variability analyses in horses” to help us in this analyses.

Reviewer 4 Report

Revision of the Case report entitled “Cartilaginous intrusion of the atrioventricular node in a Quarter horse with a high burden of second-degree AV block and 3 collapse: a case report”

The Authors presented an interesting case of an eight-year-old Quarter horse mare affected by multiple collapse episodes during rest. The clinical examination revealed a pronounced number of second-degree atrioventricular blocks and low ventricular rate. The post-mortem inspection revealed severe cartilaginous changes of the area around the aortic valve, which was intruding into the atrioventricular node in the His bundle region. This could be the reason of the slow AV conduction; however, authors themselves said that a specific diagnose could not be established and more knowledge on atrioventricular nodal disease in horses is required.

I have only some minor comments:

Line 76: add also the heart rate when second-degree AV block are present and the minimum heart rate recorded

Figure 1 ad 2: choose “paper speed” or “chart speed”

Figure 3: add the explanation of all the abbreviations

Round 2

Reviewer 3 Report

The suggestions were accepted and the paper increased the quality.